# Design and Simulation of Flexible Underwater Acoustic Sensor Based on 3D Buckling Structure

**DOI:** 10.3390/mi12121536

**Published:** 2021-12-10

**Authors:** Guochang Liu, Wenping Cao, Guojun Zhang, Zhihao Wang, Haoyu Tan, Jinwei Miao, Zhaodong Li, Wendong Zhang, Renxin Wang

**Affiliations:** State Key Laboratory of Dynamic Measurement Technology, North University of China, Taiyuan 030051, China; liuguochang@live.com (G.L.); caowenping517@163.com (W.C.); zhangguojun1977@nuc.edu.cn (G.Z.); b1906062@st.nuc.edu.cn (Z.W.); tanhy97@foxmail.com (H.T.); miaojinwei96@foxmail.com (J.M.); lzd0821@163.com (Z.L.); wdzhang@nuc.edu.cn (W.Z.)

**Keywords:** vector hydrophone, buckling, flexible, COMSOL Multiphysics

## Abstract

The exploration of marine resources has become an essential part of the development of marine strategies of various countries. MEMS vector hydrophone has great application value in the exploration of marine resources. However, existing MEMS vector hydrophones have a narrow frequency bandwidth and are based on rigid substrates, which are not easy to be bent in the array of underwater robots. This paper introduces a new type of flexible buckling crossbeam–cilium flexible MEMS vector hydrophone, arranged on a curved surface by a flexible substrate. A hydrophone model in the fluid domain was established by COMSOL Multiphysics software. A flexible hydrophone with a bandwidth of 20~4992 Hz, a sensitivity of −193.7 dB, excellent “8” character directivity, and a depth of concave point of 41.5 dB was obtained through structured data optimization. This study plays a guiding role in the manufacture and application of flexible hydrophones and sheds light on a new way of marine exploration.

## 1. Introduction

With the progress of human civilization, while the land resources are constantly exploited and utilized, the exploration of marine resources has been continuously carried out. At present, marine development has become an essential part of the strategic development of coastal countries [1]. Underwater information dissemination is faced with many problems in the process of underwater target localization and resource exploration. Electromagnetic waves decay quickly in water and have inferior penetrating force. Light waves are also limited to shorter distances due to problems of scattering and absorption in water. However, sound waves travel far underwater, allowing basic information to be transmitted [2]. The hydrophone was born in this case.

Vector hydrophones have attracted much attention due to their multidimensional vector detection performance [3,4,5]. Roskstad et al. first proposed MEMS vector hydrophone firstly in 1996 [6]. Underwater acoustic signals are mainly distributed in the low-frequency domain. Piezoresistive sensors can obtain favorable piezoresistive properties at low and even ultra-low frequencies [7]. MEMS devices are smaller, faster, have lower power consumption, and are more accurate than their macroscopic counterparts [8]. For these reasons, MEMS vector hydrophone based on piezoresistive principle has become a research hotspot. In 2006, Yang et al. designed a piezoresistive MEMS vector hydrophone by MEMS technology and piezoresistive principle [9]. In 2007, Xue and Chen creatively proposed a crossbeam–cilium MEMS piezoresistive vector hydrophone with bionic technology, and the sensitivity reached −197.7 dB (1 kHz, 2 dB = 1 V/μPa) [7]. Xue’s research group also optimized and developed a number of vector hydrophones with different cilium structures [10,11,12,13]. However, all of these hydrophones increase the area of the cilium to receive sound signals, aiming to improve the sensitivity of hydrophones but ignoring the working bandwidth and detection dimension. The fitness-wheel-shaped hydrophone proposed by Wang et al. can receive sound signals from the horizontal and vertical directions. The detection dimension of the vector hydrophone is increased to three dimensions [14]. The cup-shaped hydrophone proposed by Xu et al. updates Wang’s research and increases the sound signal’s receiving area in the horizontal direction. The working bandwidth is 20 Hz–1 kHz, and the sensitivity is −188.5 dB, which means that the hydrophone has increased sensitivity at no penalty in bandwidth [15]. Wang et al. also designed a crossbeam–cilium sensor that mimics a jellyfish-like hollow sphere and successfully applied it to detect ocean turbulence [16]. However, these MEMS vector hydroacoustic/flow sensors are based on rigid substrates and are difficult to array on curved surfaces. How to realize the flexible conformal of hydrophone is the hotspot and challenging point of research.

There are two methods of flexible conformal. One is that the flexible structure is directly attached to the bending surface. The other is that the movable 3D structure is attached to the bending surface relying on the flexible substrate. Kim et al. proposed a flexible bionic skin based on silicon nanomembrane, which can sense strain, pressure, temperature, humidity, heating, and nerve stimulation functions, with a stress sensitivity of 0.41%/kPa [17]. I. Park et al. proposed a hydrogen sensor based on flexible Pd/Si nanomembrane Schottky diode with a sensitivity (current rate of change) greater than 700 at 0.5% hydrogen concentration [18]. Mei et al. proposed a flexible transient photodetector based on wafer-compatible transferred silicon nanomembrane, with an optical flow to undercurrent ratio of 107 and response rate of 1.34 A/W (λ = 405 nm) [19]. Since the strain sensor does not need to consider the frequency response and has no movable structure, the silicon nanomembrane is directly attached to the flexible substrate. However, this kind of flexible structure is not conducive to torque transfer and is difficult to be applied to crossbeam–cilium structures. Rogers et al. proposed that silicon nanomembranes form waveform buckling structures on flexible substrates to cope with tension or compression; they prepared PN junction and MOS transistor on silicon nanomembranes and completed the buckling test, which opens a new door for the application of inorganic thin-film structures in flexible electronics [20]. With appropriate planar film design, complex 3D structures such as single-layer and multilayer spirals, annular and conical spirals, cuboid cages, flowers, scaffolding, fences, and frames can be achieved through the paper-cutting buckling process [21].

There is no report on the preparation of vector hydrophones based on flexible substrates. In this paper, a 3D flexible vector hydrophone with a movable structure designed for a paper-cutting process is simulated by COMSOL Multiphysics.

## 2. Technical Background

### 2.1. Working Principle

MEMS bionic vector hydrophone is a sensor based on piezoresistive effect. The flexible vector hydrophone consists of buckling crossbeam, central block, four-end pads, cilium, piezoresistors, and the necessary electrical connections, as shown in Figure 1 and Figure 2. The piezoresistors at both ends of the beam and metal wires form two Wheatstone bridges to realize the output of the circuit. The flexible MEMS vector hydrophone attaches the structure to the stretched flexible substrate PDMS, then releases the PDMS. Meanwhile, the four-end pads are shifted inwards for a certain distance to realize the crossbeam’s buckling. The process is (a) to (b) in Figure 2.

The acoustic signal travels to the hydrophone and is sensed by cilia. Then, the signal is transmitted to the crossbeam, which causes the stress of the crossbeam to change. At the same time, stress changes occur in the piezoresistors attached to the beam. X and Y channels in Wheatstone bridges detect vector sensors’ underwater acoustic signals, which is shown in Figure 3.

In general, the position of piezoresistors is the point on the beam where the maximum stress variation [22]. The variation in the piezoresistors are related to the stress on the beam as follows:(1)$$\frac{\Delta R}{R}=\pi_{l}\sigma_{l}+\pi_{t}\sigma_{t}$$
where *R* and Δ*R* is the resistance and resistance variation in the piezoresistor, respectively. *σ_l_* and *σ_t_* are the transverse and axial stresses on the piezoresistor, respectively. *π_l_* is the transverse piezoresistive coefficient; *π_t_* is the axial piezoresistive coefficient.

When the underwater acoustic signal is activated, the X-channel single output of Wheatstone bridge is
(2)$$U_{Ax}=\frac{\left(R_{1}+\Delta R_{1}\right)\left(R_{3}+R_{3}\right)-\left(R_{2}+\Delta R_{2}\right)\left(R_{4}+\Delta R_{4}\right)}{\left(R_{1}+\Delta R_{1}+R_{2}-\Delta R_{2}\right)\left(R_{3}+\Delta R_{3}+R_{4}-\Delta R_{4}\right)}V_{in}$$
where *U_Ax_* and *V_in_* are the output and input voltages, respectively. Δ*R*_1_~Δ*R*_4_ are the variations in piezoresistors on the Wheatstone bridge. The Y-channel single output *U_Ay_* can be gained by the same method with Δ*R*_5_~Δ*R*_8_. The piezoresistors on the beam are subjected to almost identical stress [23], so the above equation can be rewritten as
(3)$$U_{Ax}=\frac{\Delta R}{R_{0}}V_{in}$$
where *R*_0_ is the piezoresistor’s value. The Y-channel signal also outputs the acoustic signal through this principle.

### 2.2. Fabrication Process

MEMS process and the post-MEMS process can fabricate flexible buckling crossbeam–cilium hydrophones. A p-type device layer silicon-on-insulator (SOI) was used to prepare the hydrophone structure and warp structure on the PDMS. The fabrication process is shown in Figure 4.

The specific process steps corresponding to Figure 4 are as follows: (a) Reactive ion etching (RIE) SOI device layer silicon was used to form p-type piezoresistors; (b) RIE buried oxide layer (BOX) formed a crossbeam structure and silicon oxide vias; (c) boron heavy implantation was carried out to form ohmic contacts; (d) sputtering and wet corrosion of the metal was carried out to form a Wheatstone bridge; (e) SOI and wafer were adhered by paraffin and deep reactive ion etching (DRIE) handle to form silicon cilium; (f) a layer of Parylene C was deposited; (g) the structure with Parylene C was transferred; (h) oxygen plasma treatment was performed on the pads; (i) the structure was transferred to the stretched PDMS and formed covalent bonds with the PDMS; (j) Parylene C was removed; (k) PDMS was released to warp the structure; (l) nanosilver was printed, which made electrical connections with subsequent circuits at the silicon oxide vias.

### 2.3. Physical-Field Interfaces

#### 2.3.1. Solid Mechanics Interface

We chose the solid mechanics interface to describe the prestress, as well as to calibrate the sensitivity. The solid mechanics interface is a branch of structural mechanics based on solving the equation of motion and solid materials’ constitutive model. It calculates the results of displacement, stress, and strain. In this study, we added the structural mechanic’s description of inward displacement into the structure, following the prestressing state of the buckling crossbeam.

The equilibrium differential equation of solid mechanics in the transient time-varying state is
(4)$$\rho \frac{\partial^{2}\vec{u}}{\partial t^{2}}+d_{a}\frac{\partial \vec{u}}{\partial t}-\nabla \cdot \tilde{\sigma }=\vec{f}$$
where *ρ* is density, *d_a_* is damping coefficient, $\tilde{\sigma }$ is stress, $\vec{u}$ is displacement, and $\vec{f}$ is body force. The steady-state form can be rewritten as
(5)$$-\nabla \cdot \tilde{\sigma }=\vec{f}$$

The constitutive equation for a solid material according to Hooke’s law is
(6)$$S-\left(S_{0}+S_{ext}+S_{q}\right)=C:\left(\varepsilon -\left(\varepsilon_{0}-\varepsilon_{th}+\varepsilon_{hs}+\varepsilon_{pl}+\varepsilon_{cr}\right)\right)$$
where *S* is stress, *S_0_* is the prestress, *S_ext_* is external stress, *S_q_* is viscous stress, *ε* is elastic strain, *ε_0_* is prestrain, *ε_th_* thermal strain, *ε_hs_* is infiltration inflation, *ε_pl_* plastic strain, *ε_cr_* is creepage. Additionally, ***C*** is the elastic matrix, which can be expressed by Young’s modulus (*E*) and Poisson’s ratio (*v*) as follows:(7)$$D=\frac{E}{\left(1+v\right)\left(1-2v\right)}\left[\begin{array}{cccccc} 1-v & v & v & 0 & 0 & 0\\ v & 1-v & v & 0 & 0 & 0\\ v & v & 1-v & 0 & 0 & 0\\ 0 & 0 & 0 & \frac{1-2v}{2} & 0 & 0\\ 0 & 0 & 0 & 0 & \frac{1-2v}{2} & 0\\ 0 & 0 & 0 & 0 & 0 & \frac{1-2v}{2} \end{array}\right]$$

However, the equilibrium equation and the constitutive equation are not enough to solve the problem. The geometric equation of stress and displacement relationship should be added to solve the solid mechanics’ problem.
(8)$$\tilde{\varepsilon }=\frac{1}{2}\left[{\left(\nabla \vec{u}\right)}^{T}+\nabla \vec{u}\right]$$

#### 2.3.2. Pressure Acoustics–Frequency Domain Interface

Pressure acoustics–frequency domain interface is a branch of pressure acoustics. It is used to calculate the pressure change of sound wave propagation in a silent condition body. It is suitable for all frequency-domain simulations with harmonic variations in the pressure field. This interface can calculate the natural frequency, stress, strain, etc. In this study, the pressure acoustics–frequency domain interface was used to calculate the natural frequency of the hydrophone in the water area and the directivity of the signal output in X and Y channels of the hydrophone under the action of an acoustic signal.
(9)$$\nabla \left(-\frac{1}{\rho_{0}}\left(\nabla p-q\right)\right)+\frac{1}{\rho_{0}c^{2}}\frac{\partial^{2}p}{\partial t^{2}}=Q$$
where *ρ*_0_ is density, *p* is stress, **q** is dipole source, *c* is sound velocity, and **Q** is the monopole source. When the stationary state is reached, the time item can be deleted and applied to the frequency domain calculation, which is the Helmholtz equation as follows:(10)$$\nabla \left(-\frac{1}{\rho_{0}}\nabla p\right)-\frac{1}{\rho_{0}}{\left(\frac{\omega }{c}\right)}^{2}p=0$$
where *ω* is frequency, and sound propagation is dependent on frequency and sound velocity not time in the stationary state. It is linear in the frequency domain, and scanning the frequency can obtain the desired solution.

#### 2.3.3. Acoustics–Structure Interface

The multiphysics coupling here is acoustic–structure interaction. Acoustic–structure interaction usually involves solid and fluid parts. By modeling the elastic wave in solid and the pressure wave in liquid form, the trace of these two waves can be predicted, and the behavior of these two waves at the liquid–solid interface can be captured to simulate their interaction. Only complete bidirectional coupling can genuinely simulate the hydrophone operation in water.

The acoustic–structure interaction includes the fluid load on the structure during sound propagation and the structural acceleration of the fluid when the structure is moving. Its boundary condition is
(11)$$\nabla \left(-\frac{1}{\rho_{0}}\nabla p\right)-\frac{1}{\rho_{0}}{\left(\frac{\omega }{c}\right)}^{2}p=0$$
(12)$$F_{A}=p_{t}\cdot n$$
where ***u****_tt_* is structural acceleration, **n** is surface normal, *p_t_* is total acoustic pressure, *q_d_* is dipole source, and **F***_A_* is the load per unit area experienced by the structure.

## 3. Methodology

### 3.1. Geometry and Materials

The parameters are expressed as variables to make it easier to change the structure parameters in the subsequent optimization process. To build the model, first, a cuboid central block was constructed in the Cartesian coordinate system. Then, a cylinder cilium was constructed above the central block. A cuboid beam was then built on the left side of the central block, and then a cuboid pad was constructed on the left side of the beam. Next, the beam and pad were rotated every 90° to construct a crossbeam and construct a cube fluid domain. It should be emphasized that a cuboid perfectly matched layer was set to the right of the fluid domain to absorb sound waves and avoid reflection. The model is shown in Figure 5.

The crossbeam, central block, and four-end pads were made of SiO_2_, the cilia were made of Si, and the fluid domain and perfectly matched layer were made of water. Different areas of the model were controlled by their own physical-field interfaces and required different material properties. The material properties necessary for solid mechanics interfaces include Young’s modulus, Poisson’s ratio, and density. The material properties required for pressure acoustics–frequency domain interface are density and sound velocity. These material parameters are listed in Table 1.

### 3.2. Mesh

In the finite element simulation, the mesh division of the model strongly influences the simulation. Rough mesh is likely to produce significant errors in the simulation results, more evident in geometric, nonlinear, mechanical simulations [24].

In this model, the crossbeam, the central block, and the four-end pads were all rectangular structures with large slenderness ratios. These structures’ length and width had to be refined to avoid the mesh with slight skewness. In the study process, there was a gradient field with drastic changes in the beam but not in other areas. Moreover, the stress change in the beam was of utmost concern, so it was more necessary to refine it. To form a high-quality mesh, high-precision mapping and sweeping operations were applied to this part. The specific method used was to set the boundary of the structure length and width to appropriate values, form a rectangular mesh on the surface by mapping, and then sweep along the thickness direction to form a high-quality hexahedral mesh. On the other hand, cilium produced very little change in stress under load or sound pressure, compared with the crossbeam. Additionally, the cilium’s primary function was to transfer the force to the beam. Therefore, the requirement for the quality of the cilium mesh was not that high, so there was no need for a too dense mesh, which is also conducive to speeding up the calculation. The mesh of the crossbeam–cilium hydrophone is shown in Figure 6a. Conventional fluid dynamics mesh was used for water bodies. Still, in acoustic simulation, the meshing experience for wave problems is to have at least five or six second-order grid elements in each wavelength to obtain analytical waves [24]. Therefore, we divided the mesh into free tetrahedral and set its minimum mesh size as *c/f*_0_/6, where *c* is sound velocity, *f*_0_ is sound pressure maximum scanning frequency. A polynomial’s perfectly matched layers are at least 8 layers [24], so an 8-layer sweep was performed. The overall mesh is shown in Figure 6b.

The mesh quality of the fluid domain set by this method can meet the requirements of acoustic simulation. The number of meshes of hydrophone structure was 32,580, the average meshes skewness was 0.983, and the skewness of all meshes was greater than 0.53. It is worth mentioning that the skewness of all the mesh on the beam was 1, as is shown in Figure 7.

## 4. Results and Discussions

### 4.1. Natural Frequency and Stress

It should be noted that, compared with the traditional hydrophone, the island-bridge structure designed in this paper applied the sensor of prestress. The first step that needs to be studied is the buckling of the crossbeam. All the research steps, including the application of boundary load, the application of plane wave radiation, and the calculation of natural frequency, were carried out based on the first step of buckling.

The natural frequency of hydrophone also changes significantly with the change in structure size. Due to the buckling of the beam, the beam is always in a state of tension. Compared with the traditional plane stress-free device, the first-order natural frequency of the island-bridge structure is much larger than that of the stress-free state. The results of the variation in the first-order natural frequency of the hydrophone with the size in the water area are shown in Figure 8.

Natural frequency is positively correlated with beam width but negatively correlated with beam length, cilium radius, and height.

The two ends of the beam have the highest stress when the crossbeam is buckling. The two ends of the beam also experience the greatest stress variation after loading the cilium. Similar to the natural frequency, the maximum stress variation on the beam also varies with the crossbeam–cilium parameter. The maximum stress variation on the beam and the influence of the structure on the parameters are shown in Figure 9.

Maximum stress variation is negatively correlated with beam width but positively correlated with beam length, cilium radius, and cilium height. The maximum stress variation on the beam is proportional to the sensitivity, so the hydrophone’s sensitivity conflicts with the natural frequency. By analyzing the relationship between natural frequency and the maximum stress variation on the beam, the influencing factors of structure parameters were obtained, as is shown in Figure 10.

The hydrophone was fabricated with SOI, and its device layer, buried oxide layer, and handle layer thickness were limited; that is, the size of SOI limited the cilium height and the beam thickness. Moreover, the complexity of stripping on the flexible substrate also restricts the beam length and beam width [25]. Figure 10 shows the law of the influence of structure parameters on sensor performance. Considering the above, The structural parameters were determined, as shown in Table 2.

The first six natural frequencies obtained by simulation according to this size are shown in Figure 11. The working frequency band of the hydrophone is limited by its natural frequency. The resonance phenomenon appears at the first natural frequency, and the sensor measurement error occurs. Sensors must work within the first-order resonance frequency, so the upper limit of the cut-off frequency of the hydrophone is 4992 Hz. It is challenging to ensure the same resistance on the beam in the manufacturing process, so it is difficult for the hydrophone to extract ultra-low frequency signals below 20 Hz. Moreover, the ultra-low frequency signals in the water are mixed with a considerable amount of noise, and there is considerable low-frequency noise in the subsequent signal processing circuit, so the sensor is not sensitive to ultra-low frequency signals below 20 Hz. Therefore, the bandwidth of the sensor is 20–4992 Hz.

### 4.2. Sensitivity

The buckling process of the crossbeam is very nonlinear. The typical direct solver is easy to produce non-convergent results, which brings great difficulty to stress simulation. In this study, the full coupling form adopts the constant Newton nonlinear solution method, in which the constant damping factor 1 was used. The four-end pads will be shifted inward by 0.1 um at a time in the form of an auxiliary scan. At each iteration, the new Jacobian of all iterations of Newton’s method was calculated, which is more beneficial to the convergence of the operation. The crossbeam was buckled by the constant Newton method. The beam’s stress contours and curve are shown in Figure 12a and Figure 13a, respectively.

After applying 1 Pa stress in the positive X direction, the cilium transfers the stress to the buckling crossbeam, as is shown in Figure 12b. The simulation results show that the maximum stress variation on the beam is distributed near the end of the beam. The variation of up to 31.6 kPa is generated at the end of the two beams in the X direction, as shown in Figure 13b. Through the comparison calibration method [7], the sensitivity of the hydrophone in this study is
(13)$$S_{f}=-197.9+20\;\mathrm{lg}\left(\frac{p_{F}}{p_{CVH}}\right)$$
where *S_f_* is sensitivity, *p_F_* = 31.6 kPa is maximum stress variation in the flexible hydrophone, and *p_CVH_* = 20 kPa is maximum stress of traditional hydrophone. The sensitivity of the flexible hydrophone is −193.9 dB.

### 4.3. Directivity

The most significant difference between a vector hydrophone and a scalar hydrophone is the ability to identify directions. When sound waves in different directions act on the hydrophone, the output of the two signals of the crossbeam is different. The direction of sound propagation is identified by the difference in the output size of the two signals.

For this study, 1 Pa plane wave radiation was applied to the positive X direction of the hydrophone on one side of the fluid domain. The cilium was disturbed by the influence of sound pressure so that the stress on the beam would change. Then, the plane wave radiation was rotated 360° with a step of 5° to obtain different stress variations in the X and Y channel. The stress was normalized by Equation (14).
(14)$$L_{\theta }=20\mathrm{lg}\left(\frac{p_{x}}{p_{xmax}}\right)+\left|20\;\mathrm{lg}\left(\frac{p_{xmin}}{p_{xmax}}\right)\right|$$
where *L_θ_* is radial length, *p_x_* is stress variation, and *p_xmax_* and *p_xmin_* are maximum and minimum stress variations. Figure 14 shows the directivity of the hydrophone at the acoustic frequency of 1 kHz. The depth of the concave point can reach 45.1 dB. The “8” character directivity is excellent.

## 5. Conclusions

In this paper, a new type of flexible vector hydrophone with wide bandwidth and high sensitivity based on buckling crossbeam–cilium was designed. Based on COMSOL Multiphysics simulation analysis, this hydrophone’s actual measurement frequency range is 20~4992 Hz, which can realize the full deep-sea ultra-wide bandwidth detection. The sound pressure gradient sensitivity reaches −193.9 dB, which is 4 dB higher, compared with traditional hydrophones. It has a good directivity of “8” character and a depth of concave point reach 45.1 dB. It has excellent two-dimensional vector detection performance. The hydrophone can be attached to different surfaces through the flexible substrate and buckling structure to have a broader range of applications. This study lays a theoretical foundation for the manufacture and application of the new flexible vector hydrophone. It is expected to be manufactured shortly and applied in underwater robots, airborne sonar buoys, and other underwater detectors. It provides new opportunities and possibilities for the exploration of the marine environment and resources.

## Figures and Tables

**Figure 1 micromachines-12-01536-f001:**
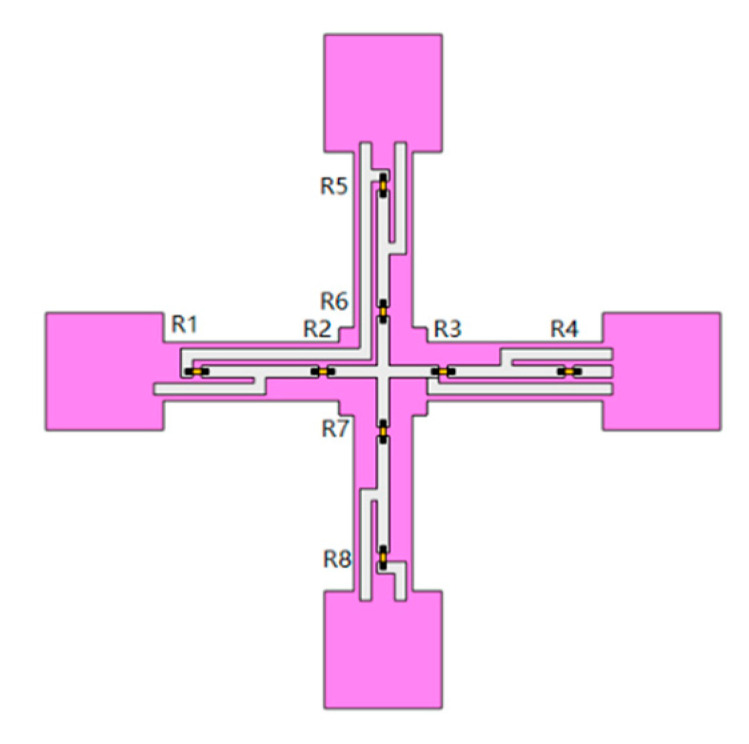
Diagram of piezoresistors on crossbeam.

**Figure 2 micromachines-12-01536-f002:**
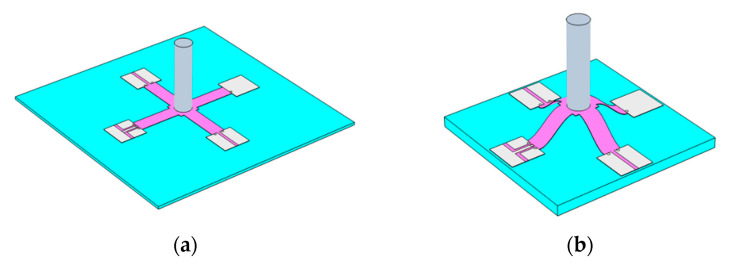
Structure on flexible substrate: (**a**) structure attached to stretched flexible substrate; (**b**) buckling structure on the released flexible substrate.

**Figure 3 micromachines-12-01536-f003:**
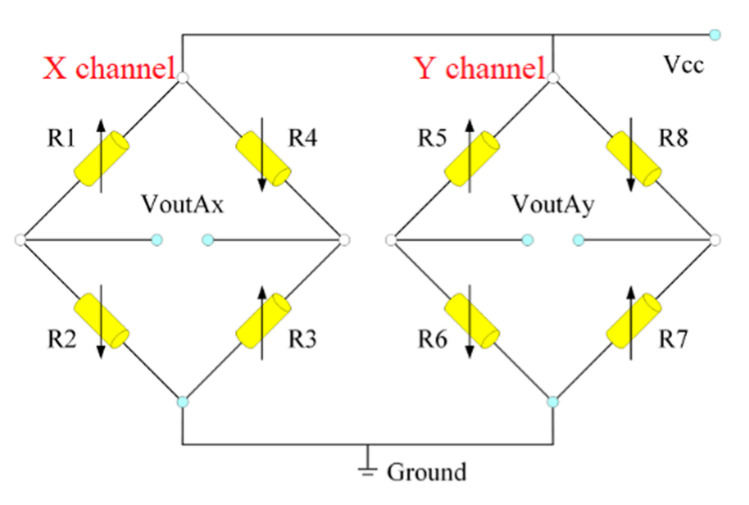
Wheatstone bridge.

**Figure 4 micromachines-12-01536-f004:**
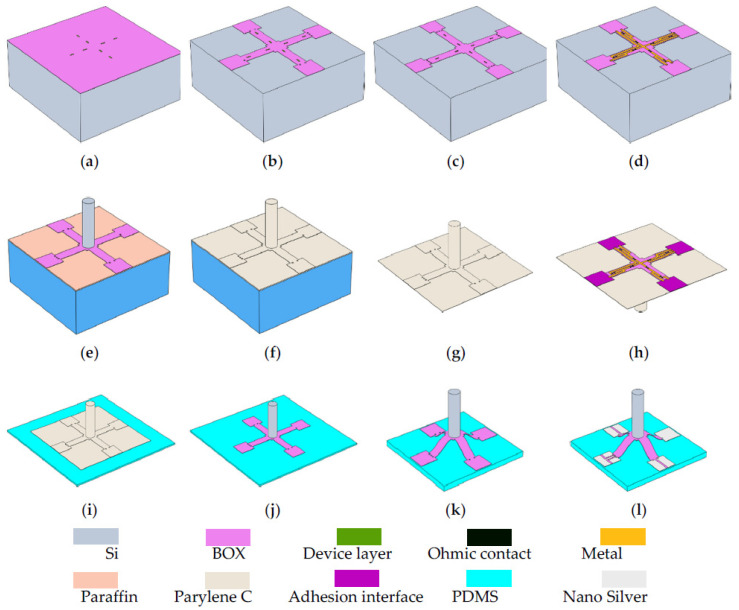
Sketch of the microfabrication process: (**a**) etch device layer; (**b**) etch BOX layer; (**c**) boron implantation; (**d**) sputter and graphical metal; (**e**) transfer to wafer and etch handle; (**f**) deposition Parylene C; (**g**) lift-off from water; (**h**) selective plasma treatment; (**i**) adsorption on the stretched PDMS; (**j**) remove Parylene C; (**k**) release PDMS and implement buckling; (**l**) electrical connections.

**Figure 5 micromachines-12-01536-f005:**
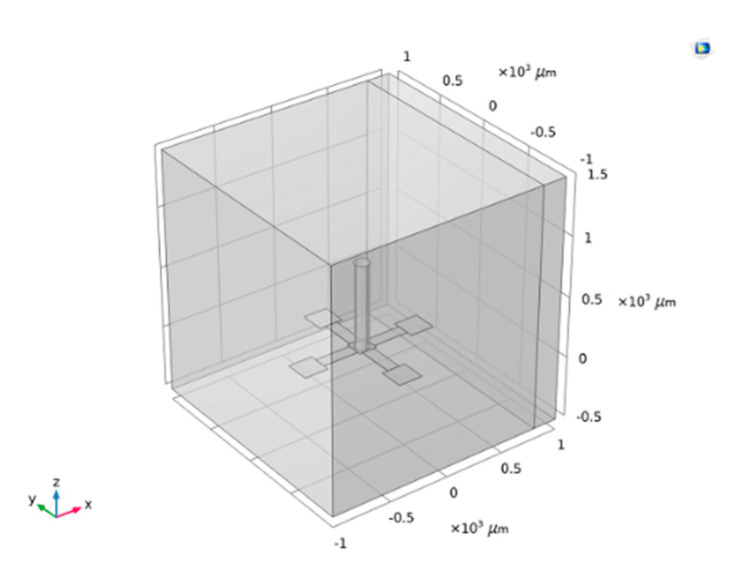
Geometry model.

**Figure 6 micromachines-12-01536-f006:**
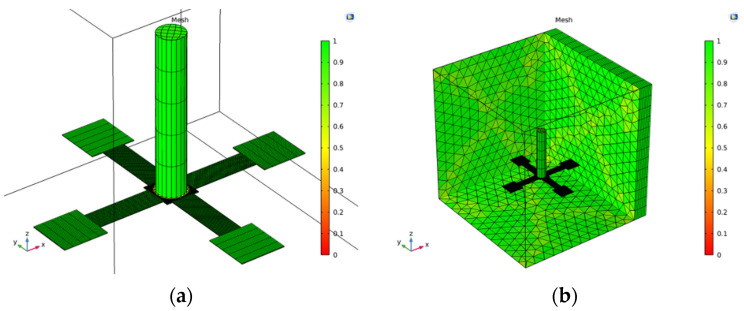
Mesh: (**a**) mesh of crossbeam–cilium; (**b**) overall mesh.

**Figure 7 micromachines-12-01536-f007:**
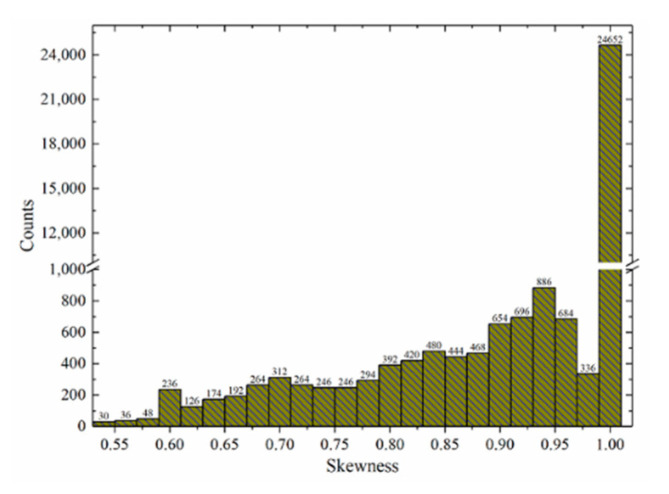
Mesh quality histogram.

**Figure 8 micromachines-12-01536-f008:**
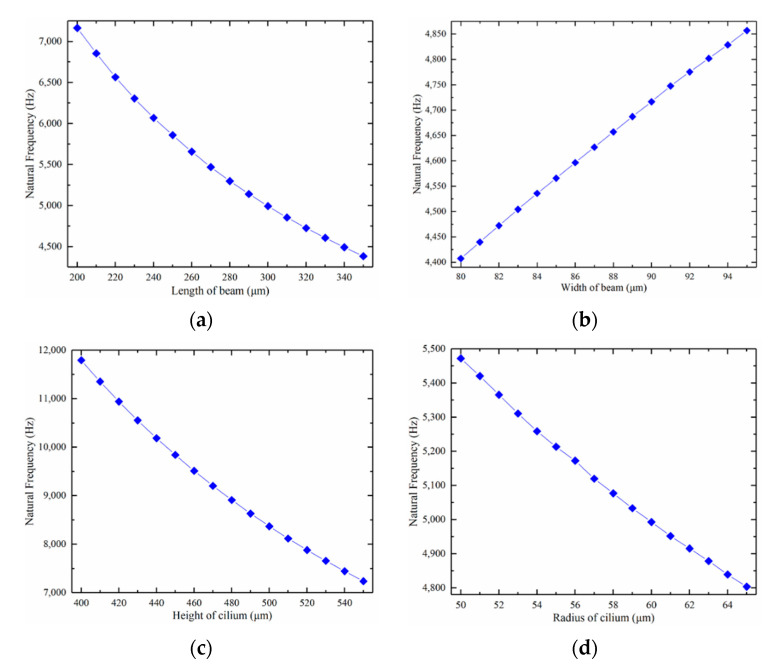
The impact of different structural parameters on the natural frequency: (**a**) beam length’s impact; (**b**) beam width’s impact; (**c**) cilium height’s impact; (**d**) cilium radius’ impact.

**Figure 9 micromachines-12-01536-f009:**
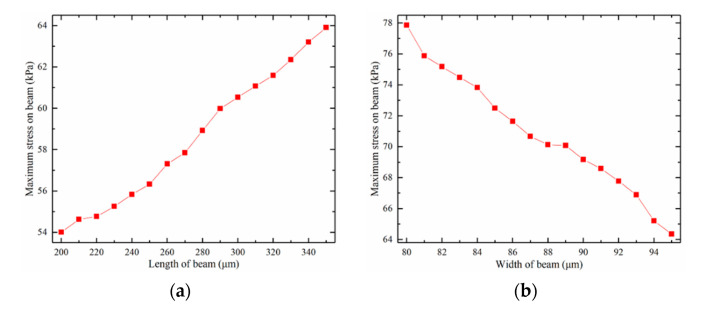
The impact of different structural parameters on the max-stress variation: (**a**) beam length’s impact; (**b**) beam width’s impact; (**c**) cilium height’s impact; (**d**) cilium radius’ impact.

**Figure 10 micromachines-12-01536-f010:**
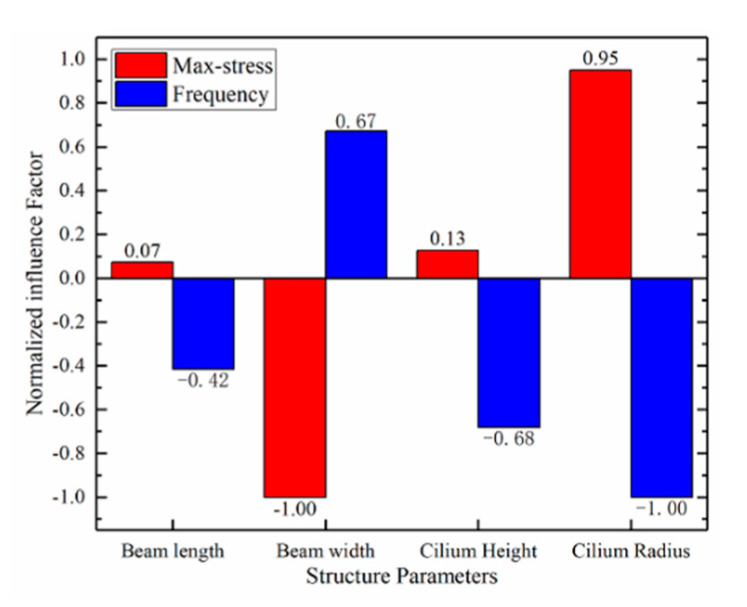
Influence factor of each parameter.

**Figure 11 micromachines-12-01536-f011:**
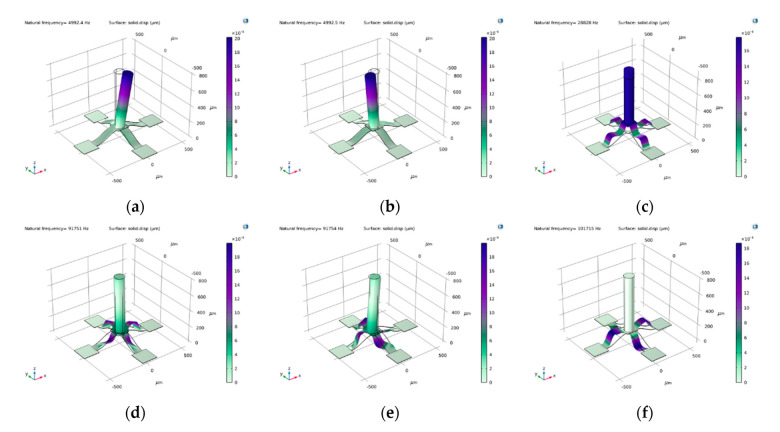
The natural frequency of microstructure: (**a**) first natural frequency; (**b**) second natural frequency; (**c**) third natural frequency; (**d**) fourth natural frequency; (**e**) fifth natural frequency; (**f**) sixth natural frequency.

**Figure 12 micromachines-12-01536-f012:**
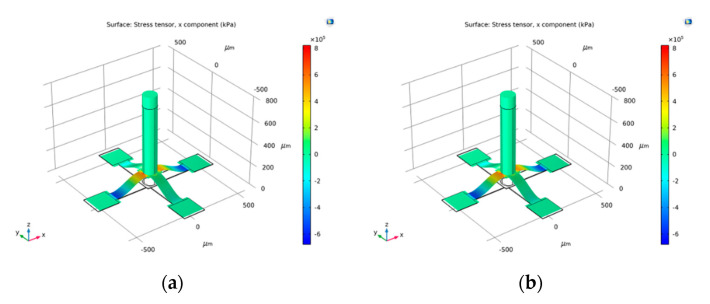
Stress tensor contours: (**a**) buckling stress contours; (**b**) post-buckling stress contours.

**Figure 13 micromachines-12-01536-f013:**
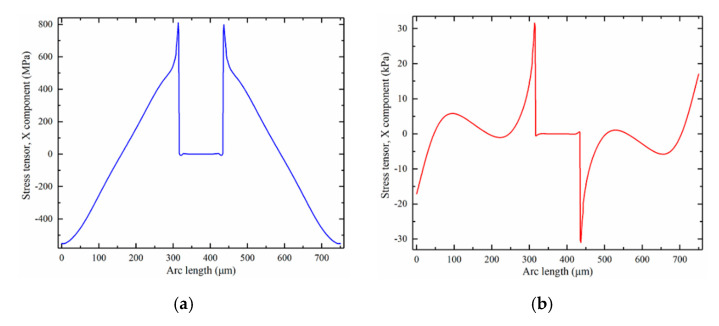
Stress tensor curve of beam: (**a**) buckling stress curve; (**b**) variation in post-buckling stress curve and buckling stress.

**Figure 14 micromachines-12-01536-f014:**
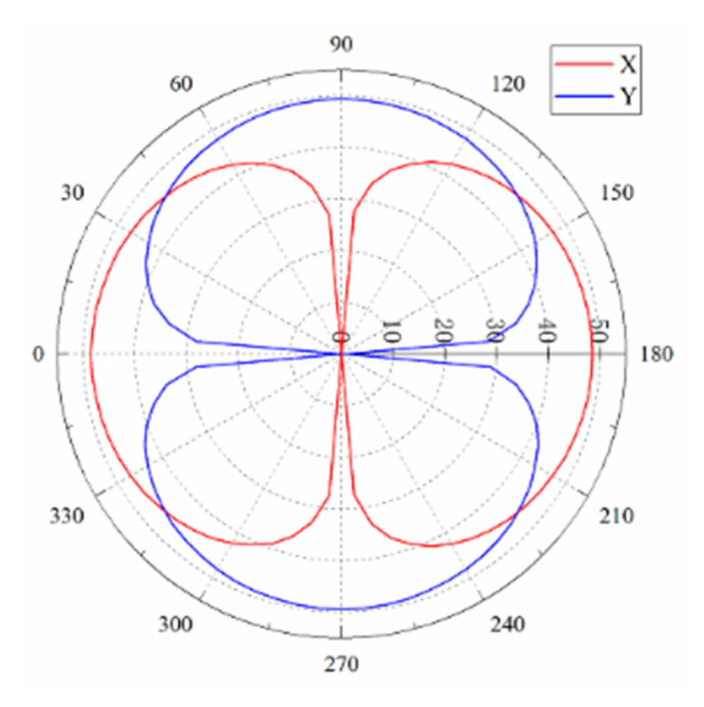
The “8” character directivity diagram.

**Table 1 micromachines-12-01536-t001:** Material properties.

Material	Young’s Modulus (GPa)	Poisson’s Ratio	Density (kg/m^3^)	Sound Velocity (m/s)
Si	130	0.27	2329	/
SiO_2_	70	0.2	2200	/

**Table 2 micromachines-12-01536-t002:** Values of structure parameters.

Structure Name	Parameter (μm)
Length of the crossbeam	300
Width of the crossbeam	100
Thickness of the crossbeam	3
Length of the pad’s side	200
Length of the central block’s side	150
Height of the cilium	700
Radius of the cilium	60

## Data Availability

Not applicable.

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
