# Peer review of "Design and Simulation of Flexible Underwater Acoustic Sensor Based on 3D Buckling Structure"

_micromachines, 2021, doi:10.3390/mi12121536_

Round 1

Reviewer 1 Report

In this work, the authors propose a novel flexible vector hydrophone based on buckling crossbeam-cilium. The idea is not turned into reality, but the principle and simulation works make it reasonable and understandable. Though there are some grammar mistakes and the current manuscript still needs to be polished, in my opinion, the paper can be considered to be published in Micromachines after addressing the below issues.

1. The authors analyze the sensitivity and directivity, but there is no direct discussion about the bandwidth in the manuscript. So how do you get the bandwidth 20-4992Hz?

2. The title of 2.2.3 should be Acoustics-Structure Interaction?

3. In part 3.2, the authors indicate that the matching layer for a polynomial is at least 8 layers, so 8 layers are chosen. How about more layers? The simulation performance could be better if more layers are used in the simulation?

4. The proposed hydrophone has good frequency bandwidth, sensitivity, and directivity, even compared with those traditional hydrophones. I understand those numbers are given from the simulation analysis, but it is better to analyze the physical causes, i.e. what parameters or structures give this improvement?

Reviewer 2 Report

  1. Line 109 mentions channel X: Please indicate in the figure or explain what it means.
  2. Line 104 has a text format line.
  3. It is advisable to mark R1, R2,.. to R8 in Figures 1 and/or 2.
  4. Have the authors studied what would happen if there is an asymmetry between the arms? For instance, while fabrication it often happens that not all arms are identical.
  5. With reference to Line 119, do the authors imply that Uax = Uay.
  6. Explain what sensor of prestress means in Line 223.
  7. Can the authors analytically derive an equation for the structure and thereby explain the dependency of the device’s response (Fig. 7-9) to varying device geometries.
  8. The authors present trivial information in Figure 7-9 which can be essentially a couple of figures to explain the effect of device geometry on device performance. It will be more insightful if there can be discussions on how the device dimensions affect the Wheatstone bridge configuration and its response.
  9. The authors show 6 mode shapes in Figure 10, but their explanation is missing in the main text. Please highlight the point/reason for this study.
  10. Figure 11 a and b appear to be the same. Please explain why?
  11. Figures 12 and 11 do not agree. And again, these two figures are co-related and can be combined as a single figure.
  12. Comment on the huge variations of stress for Figure 12 (a) and (b) (close to 30X difference).
  13. Did the authors study the effect of medium and damping associated with AVS for directivity and bandwidth analysis?
  14. What is the minimum change in deflection that can be detected by the change in resistance for the current design?

Reviewer 3 Report

This manuscript demonstrated the design and simulation process of a MEMS piezoresistive hydrophone without any experimental data. The theoretical analysis is based on traditional mature theories, the authors didn't propose anything new. The simulations are also commonly used by other people and it's not necessary to illustrate the simulations so much. If the authors could supply the fabrication process, the fabricated devices, and the experimental data, then this manuscript would be better. But the current version is not acceptable.

Author Response

Thank you for taking the time to review my paper.

In this work, we propose a novel flexible vector hydrophone based on buckling crossbeam-cilium. The idea is not turned into reality, but the principle and simulation works make it reasonable and understandable.

No one has combined buckling structures with underwater sound. The hydrophone thus designed has high bandwidth, sensitivity and directivity. The hydrophone can be attached to different surfaces through the flexible substrate and buck-ling structure to have a broader range of applications. This study lays a theoretical foun-dation for the manufacture and application of the new flexible vector hydrophone. It is expected to be manufactured shortly and applied in underwater robots, airborne sonar buoys and other underwater detectors. It provides new opportunities and possibilities for the exploration of the marine environment and resources.

We appreciate for reviewers’ warm work earnestly and hope that the correction will meet with approval. Once again, thank you very much for your comments.

Round 2

Reviewer 3 Report

Although this manuscript provides new opportunities and possibilities for the combination of buckling structures and underwater sound detection. It will be better if the authors could provide a fabrication process for the proposed device.
